First occurrence of the enigmatic peccaries Mylohyus elmorei and Prosthennops serus from the Appalachians: latest Hemphillian to Early Blancan of Gray Fossil Site, Tennessee

Doughty Evan M. emdoughty@g.ucla.edu 1
Wallace Steven C. 2
Schubert Blaine W. 2
Lyon Lauren M. 2 3
1 Department of Ecology and Evolutionary Biology, University of California, Los Angeles , Los Angeles , CA , United States of America
2 Department of Geosciences and Don Sundquist Center of Excellence in Paleontology, East Tennessee State University , Johnson City , TN , United States of America
3 Department of Ecology and Evolutionary Biology, University of Tennessee—Knoxville , Knoxville , TN , United States of America
Farke Andrew
Electronic publication date: 2018 Nov 30
Publication date: 2018
Volume: 6
Electronic Location ID: e5926
Received 2018 Aug 15; Accepted 2018 Oct 13
Copyright: ©2018 Doughty et al.
Copyright year: 2018
Copyright holder: Doughty et al.
License: This is an open access article distributed under the terms of the Creative Commons Attribution License, which permits unrestricted use, distribution, reproduction and adaptation in any medium and for any purpose provided that it is properly attributed. For attribution, the original author(s), title, publication source (PeerJ) and either DOI or URL of the article must be cited.
License URL: https://creativecommons.org/licenses/by/4.0/

Keywords: Gray Fossil Site, Latest Hemphillian, Early blancan, Artiodactyla, Tayassudiae, Tayassuinae, Paleontology, Niche partitioning, Intraspecific variation, Hypoconule/hypoconulid complex

Funding: National Science Foundation EAR-0958985 CSBR 1203222 Don Sundquist Center of Excellence in Paleontology Research and Sponsored Programs Department of Geosciences This research was supported by the National Science Foundation (EAR-0958985), and the following divisions of ETSU: Don Sundquist Center of Excellence in Paleontology, Research and Sponsored Programs, and the Department of Geosciences. University of Florida specimens were photographed under NSF grant CSBR 1203222. There was no additional external funding received for this study. The funders had no role in study design, data collection and analysis, decision to publish, or preparation of the manuscript.

==============================
Two peccary species, Mylohyus elmorei and Prosthennops serus are described from the medium-bodied fauna of the Gray Fossil Site (GFS) of northeastern Tennessee. This site, recognized as an oak-hickory forest, is latest Hemphillian or earliest Blancan based on mammalian biochronology, with an estimated age of 4.9–4.5 Ma. The GFS represents the only site outside the Palmetto Fauna of Florida with M. elmorei, greatly expanding the species range north over 920 km, well into the Appalachian region. This is also the first Appalachian occurrence of the relatively widespread P. serus. Our understanding of intraspecific variation for both M. elmorei and P. serus is expanded due to morphological and proportional differences found in cranial and dental material from the GFS, Tyner Farm locality, Palmetto Fauna, and within the literature. The GFS M. elmorei material represents the most complete mandible and second cranium for the species, and preserve intraspecific variation in the length of the diastema, dental proportions, and the complexity of the cuspules of the hypoconulid complex. Similarly, mandibular material from the GFS for P. serus exhibited larger dentitions and a greater degree of robustness than currently recognized for the species.

Introduction

Tayassuidae, a family of pig-like artiodactyls endemic to the New World, is geographically and temporally widespread (Wright, 1998). East of the Mississippi River, however, late Hemphillian to early Blancan faunas are rare. The Palmetto Fauna of Florida is similar in age to Gray Fossil Site (GFS), and is represented by an aggregation of multiple mine localities (e.g., Fort Green Mine, Palmetto Mine, Payne Creek, Saddle Creek Mine, and South Pierce quarries) within the Central Florida Phosphate District of Polk, Hillsborough, Manatee, and Hardee counties (Fig. 1) (Wright & Webb, 1984; Wright, 1989; Hulbert, 2001; Webb et al., 2008). Outside of the Palmetto Fauna, the eastern record is represented by the Pipe Creek paleosinkhole of Indiana (Hh4) (Farlow et al., 2001; Prothero & Sheets, 2013), the Mauville local fauna of Alabama (Hh2) (Hulbert & Whitmore, 2006), and the Tyner Farm locality of Florida (late Hh1 to early Hh2) (Hulbert & Whitmore, 2006; Hulbert, Morgan & Kerner, 2009a). Specifically, the Tyner Farm and Mauville local faunas both chiefly exhibit Prosthennops serus (Hulbert & Whitmore, 2006; Hulbert, Morgan & Kerner, 2009a) whereas the Pipe Creek fauna is attributed by Prothero & Sheets (2013) to include Protherohyus brachydontus (=Catagonus brachydontus; see Parisi Dutra et al. (2017) and Platygonus pollenae. Tayassuids are also identified within the fauna present at the Gray Fossil Site (GFS), with Parmalee et al. (2002) suggesting cf. Protherohyus sp. However, based on the fragmentary nature of the material recovered at that time, subsequent workers identified GFS peccaries to the family level only (e.g., Wallace & Wang, 2004; DeSantis & Wallace, 2008).

Figure 1 Eastern to southeast United States (US) showing locations of the Hemphillian sites bearing peccary material (A).

The Gray Fossil Site is located in north-central Washington County (B), TN at 36.5°N and 82.5°W. Mylohyus elmorei was only known from various localities within the Phosphate Mines (grey) of the Palmetto Fauna (C) (Wright & Webb, 1984; Wright, 1991; Wright, 1998). Wright (1998) reports material with affinity to Mylohyus longirostris from the Mixon’s Bone Bed local fauna of northern Florida; however, no specimens are directly listed to verify this claim. Prosthennops serus is identified within the Mauville Fauna of southern Alabama (Hulbert & Whitmore, 2006) and the Tyner Farm locality of northern Florida (Hulbert, Morgan & Kerner, 2009a). Protherohyus brachydontus (=Catagonus brachydontus), despite being widespread in the western US and Mexico, is currently only recognized in the Palmetto Fauna (Wright, 1989; Wright, 1991; Wright, 1998) and the Pipe Creek paleosinkhole of Indiana (Prothero & Sheets, 2013) in the eastern US.

The GFS, located in eastern Tennessee (Fig. 1), is one of the few localities in the eastern United States that represents the latest Hemphillian (Hh4) to earliest Blancan (Wallace & Wang, 2004; Samuels, Bredehoeft & Wallace, 2018). An oak-hickory forest surrounded the approximately 2.6–3.5 ha lacustrine deposit, which has a maximum depth of 42 m; the GFS is comprised of up to eleven paleosinkholes within the Cambrian to Ordovician dolostone of the Knox Group (Shunk, Driese & Clark, 2006; Whitelaw et al., 2008). For a full review of the geology of GFS see Shunk, Driese & Clark (2006) and Shunk et al. (2009). Early descriptions from the GFS constrain the site to 7 to 4.5 Ma, based on the presence of Teleoceras and Plionarctos, but a recent description of Gulo sandorus from the site included a list of additional fauna which suggest an lower age limit of approximately 4.9 Ma (Wallace & Wang, 2004; Samuels, Bredehoeft & Wallace, 2018). This diverse fauna and flora from this site appears to suggest that the oak-hickory forest surrounding the lacustrine system was a refugium for otherwise disappearing taxa due to the spread of grasslands throughout other regions of the United States (Wallace & Wang, 2004; DeSantis & Wallace, 2008). Despite bearing taxa with Asiatic affinities (e.g., Wallace & Wang, 2004; Liu & Jacques, 2010; Ochoa et al., 2012), much of the fauna of the GFS exhibits great similarity to that of the Palmetto Fauna of Florida (Hulbert et al., 2009b; Bourque & Schubert, 2015). This includes Tapirus polkensis, Teleoceras sp., Plionarctos sp., and Alligator sp. (Wallace & Wang, 2004; Webb et al., 2008; Short, 2013). These similarities extend to the previously unstudied GFS peccary material and the multiple tayassuid taxa recognized within the Palmetto Fauna (Wright & Webb, 1984; Wright, 1989; Wright, 1998; Hulbert, Morgan & Kerner, 2009a).

Figure 2 Measurements for tayassuid cranial (A and B) and mandibular (C and D) material (ETVP 17584, an adult Pecari tajacu).

Measurements are modified from Von den Driesch (1976) with the abbreviation PCD representing postcanine diastema.

Considering the above, analysis of the GFS tayassuid material provides a unique opportunity to better understand the latest Hemphillian to earliest Blancan of the Appalachian region and its relation to other similarly aged sites within eastern North America. Here we report the first occurrence of Mylohyus elmorei outside the Palmetto Fauna of Florida, recognize Prosthennops serus within Appalachia, and discuss the implications of multiple tayassuid specimens occurring at the GFS and the region. Additional tayassuid material is recognized within GFS, however, more material is required.

Methods and Materials

Linear measurements (mm) follow Von den Driesch (1976), whereas nomenclature regarding skeletal morphology follows Woodburne (1968), Sisson & Grossman (1975), Wright (1989), Wright (1991) and Wright (1998) (Fig. 2). Dental measurements and nomenclature were adapted from Wright & Webb (1984) and Wright (1991) (Fig. 3). Images of specimens within the University of Florida Museum of Natural History collections were taken with a Nikon d5100 camera using a Nikon AF-S Micro-NIKKOR 60 mm f/2.8G ED lens and are available on the FLMNH website (www.flmnh.ufl.edu/vertpaleo-search/) through NSF grant CSBR 1203222 (S Moran, pers. comm., 2015). GFS specimens were photographed using a Canon EOS Rebel Xsi camera and tripod or MK DigitalDirect Photo-eBox Plus Digital Lighting System. All images were edited using GIMP 2.0, Inkscape 0.91, and Adobe Photoshop CS2 and CS5.

Specimen Repositories—ETMNH, East Tennessee State University Museum of Natural History—Fossil Collections, Gray, Tennessee; ETVP, East Tennessee State University Museum of Natural History—Comparative Collection, Johnson City, Tennessee; UF, Division of Vertebrate Paleontology, Florida Museum of Natural History, University of Florida, Gainesville, Florida; UF/TRO, Timberlane Research Organization, Lake Wales, Florida (part of the John Waldrop Collection now housed at the Division of Vertebrate Paleontology, Florida Museum of Natural History, Gainesville, Florida).

Results

Systematic Paleontology

Class MAMMALIA Linnaeus, 1758	
Order ARTIODACTYLA Owen, 1848	
Family TAYASSUIDAE Palmer, 1897	
Subfamily TAYASSUINAE Palmer, 1897	
Genus MylohyusCope, 1889	
Mylohyus elmorei (White, 1942) Wright & Webb, 1984	

Holotype—MCZ 3805: partial L. ramus with p2-m3.

Referred Specimens—ETMNH 7279: L. M2 with partial maxilla; ETMNH 8046: reconstructed partial maxilla with L. and R. P2-M3, partial mandible with L. p3-m3 and R. p2-m3; ETMNH 17219: partial L. p3; ETMNH 19281: L. m2; UF/TRO 440, UF 203540: isolated R. M3. Additional referred specimens listed by White (1942) and Wright & Webb (1984) (Table S1).

Figure 3 Dental measurements for both upper and lower dentition (A) follow Wright & Webb (1984) and Wright (1991) with the addition the transverse measurement across the hypoconule/hypoconulid complex.

Abbreviations: APO, greatest anteroposterior length taken along the midline of the occlusal surface of the tooth; AT, greatest transverse width of the trigon/trigonid cusps; PT, greatest transverse width of the talon/talonid cusps; HT, greatest transverse width of the hypoconulid complex. Dental nomenclature (B–F) is modified from Wright (1989), Wright (1991) and Wright (1998) in regards to the posterior heel of upper and lower m3. This accessory cusp-bearing, posteriorly-oriented extension of the M3 and m3’s talon/talonid is not referred to in a unified manner within the literature but is typically referred to as the “posterior lobe”, “posterior heel”, or “heel” ( Matthew, 1924; Kinsey, 1974; Schultz & Martin, 1975; Dalquest & Mooser, 1980; Wright & Webb, 1984; Wright, 1989; Wright, 1991). Despite indicating a general location on the M3/m3, these terms are undescriptive as they do little to describe the composition of the feature, which bears the hypoconule/hypoconulid and a variable number of accessory cusps. Consequently, the terms hypoconule complex and hypoconulid complex are used herein to better describe both the placement and composition of this feature on the upper and lower dentition, respectively. General morphology of the hypoconulid complex is provided for Prosthennops serus (D) and Mylohus elmorei (E); however, it should be indicated that there is notable inter- and intraspecific variation in the number of accessory cuspules on the hypoconulid complex.

Description—Cranium of ETMNH 8046 exhibits a partial maxilla with the left portion of the laterally convex rostrum extending from the P2 to the anterior margin of the canine buttress (Fig. 4). Buttress exhibits an irregular shape; bearing both triangular and hemispherical outlines in dorsal and lateral views, respectively. A deep, anteroventrally tapering concavity separates the buttress and the rest of the rostrum. Concavity is bordered laterally by a thin anteromedial to posterolaterally oriented crest that increases in robustness posterodorsally until merger with the dorsal apex of the buttress. Occlusal surface of the anterior buttress is comprised of a triangular patch of relatively flat cortical bone that is separated from the bulbous, inflated lateral margins of the buttress by a shallow ridge. Left canine alveolus is intact and ellipsoid in outline but canine is not present. Post-canine diastema between the posterior boundary of the canine alveoli and P2 is long—approximately 110% of total cheek tooth series length (Table 1)—and contains a shallow diastemal crest that trends its full length. Rostrum exhibits a laterally extending crest along the dorsal surface that is the origination of pneumatic zygoma. Crest is bulbous and inflated along its anterodorsal margin whereas the inferior boundary is marked by posteroventrally trending curvature of the maxillary bone. No internal structure of this portion of the zygoma is preserved; however, a deep sinuous depression is present posteromedial of the crest. A shallow to moderate supraorbital sulci trends the length of the dorsolateral surface of the rostrum, originating posteromedial to the lateral crest and terminating anterior and dorsal of the left canine buttress. Ventral surface of rostrum exhibits a palatine sulcus that extends medial of the P2 to the canine buttresses. The remaining medial portion of the palate posterior of the P2 is reconstructed with the posterior portion of the sulci lacking. Maxillopalatine region along the dorsal surface of the palate is sinuous. Within the maxillopalatine labyrinth, the thin—approximately 1.52 mm width—nasal septum diverges approximately 7.5 mm anterior of the P2. Internal surface of maxilla dorsal and anterior of the P2 appears to exhibit thin, shallow anteroposteriorly trending sulci. Nasal passage is incomplete with the medially projecting remnants indicating a posteriorly constricting tubular profile that trends posterodorsally from medial of the canine buttress to medial of the origination of the zygoma, dorsal to the chambers of the maxillopalatine labyrinth. Vomeroethmoid chamber is directly ventral of the nasal passage and lateral of the nasal septum. Lateral expansion of the chamber is evident posterior of the canine buttresses due the lateral bulging of the cortical bone of the maxilla, with the external surface being convex whereas the interior surface is comprised of a moderate elliptical depression. Medial surface of the rostral cortical bone is pockmarked by numerous intersecting sulci of very shallow depths. Further analysis of the maxillopalatine labyrinth as described by Wright (1991) is not possible due to the lack of preservation.

Figure 4 Gray Fossil Site tayassuid material assigned to Mylohyus elmorei, ETMNH 8046.

This partial cranium exhibits an intact left canine alveolus and right and left P2-M3. Views: (A) Lateral; (B) occlusal; (C) medial. Image is in grayscale to prevent morphologies from being obscured due to coloration.

Mandible of ETMNH 8046 exhibits an elongate, gracile condition and is mostly complete but lacking the anterior margin of the symphysis, right mandibular condyle, and right coronoid process (Fig. 5). Canine alveoli and the anterior margin of symphysis are missing. Despite being incomplete, the symphysis is relatively gracile and elongate with a moderate to deep, medially positioned spout-like concavity along its anteroposterior length that bears similarities with the mandibular spout present in ground sloths (e.g., Mcdonald & De Muizon (2002); De Muizon et al. (2003)). This mandibular concavity is laterally bounded by raised ridges that trend posteriorly and then dorsoposteriorly until the base of the p2. A single mental foramen is evident ventral to the trigon of the p4 on the labial surface of the right rami. Two genial pits are positioned within a shallow laterally trending genial fossa along the posterior surface of the symphysis medial to the rami. A shallow transverse ridge trends along the ventral-most edge of the symphasis ventral to the genial fossa. Distance between anterior edge of the p2 to the posterior edge of the symphysis is ∼43.5 mm. Rami are mediolaterally gracile with a relatively consistent depth along the cheek tooth series, however, the region in contact with the cheek teeth is medially inflated relative to the ventral margin of the rami (Table 2). Ventral surface of the rami retains a relatively similar width leading to the development of a shallow digastric fossa and submandibular fossa between the p2 and m3 which opens posteriorly into the shallow pterygoid fossa. Ventral to the m3, the left mandibular foramen, despite being damaged, appears to be ellipsoid in profile as it opens into a moderate to shallow, anteroposteriorly trending mylohyoid groove. A small mental foramen is positioned ventral to the anterior cusps of the p4 along the labial surface of the right rami. Coronoid process exhibits a triangular outline with a shallow to moderately deep masseteric fossa. Angle originates approximately ventral to the posterior margin of the m3 and exhibits a shallow pterygoid fossa that is bounded posteroventrally by a shallow ridge.

Table 1 Measurements (mm) of the upper dentition and cranium of Mylohyus elmorei.

							ETMNH 8046	UF 12265			
		N	X ¯	Range	σ2	σ	Left	Right	Average	Left	Right	Average	UF
203540	UF/TRO
440	
Length P2-M3	2	96.45	94.91–98.00	2.38	1.54	97.68	98.31	98.00		94.91				
Length P2-P4	2	37.40	37.35–37.46	0.00	0.06	36.78	37.91	37.35		37.46				
Length M1-M3	2	59.20	57.20–61.20	4.01	2.00	61.27	61.13	61.20		57.20				
Post Canine Diastema	2	101.59	95.68–107.51	34.95	5.91	107.51				95.68*				
% Length of PCD/ Length P2-M3	2	105	101–110	0.00	5	110				101				
Height of Canine Buttress	1	51.21						51.21						
Canine	APA	1	22.52	 	 	 	 	 	22.52					
Transverse	1	15.60	 	 	 			15.60					
P2	APO	2	10.34	10.31–10.36	0.00	0.02	10.37	10.25	10.31		10.36				
AT	2	9.50	9.32–9.69	0.03	0.19	9.69	9.69	9.69		9.32				
PT	2	9.47	9.32–9.63	0.02	0.15	9.67	9.58	9.63		9.32				
P3	APO	2	12.44	12.36–12.51	0.01	0.08	12.49	12.54	12.51		12.36				
AT	2	11.39	11.36–11.42	0.00	0.03	11.39	11.45	11.42		11.36				
PT	2	12.24	12.20–12.28	0.00	0.04	12.09	12.31	12.20		12.28				
P4	APO	2	14.31	14.25–14.38	0.00	0.07	14.28	14.48	14.38		14.25				
AT	2	12.69	12.22–13.17	0.23	0.48	12.01	12.42	12.22		13.17				
PT	2	14.26	13.98–14.54	0.08	0.28	13.68	14.29	13.98		14.54				
M1	APO	2	17.63	17.09–18.17	0.29	0.54	18.33	18.00	18.17		17.09				
AT	2	16.16	16.11–16.20	0.00	0.05	16.09	16.13	16.11		16.20				
PT	2	16.38	16.33–16.43	0.00	0.05	16.44	16.41	16.43		16.33				
M2	APO	2	19.33	19.07–19.58	0.07	0.26	19.62	19.54	19.58	19.16	18.98	19.07			
AT	2	17.57	17.30–17.84	0.07	0.27	17.88	17.81	17.84	17.23	17.38	17.30			
PT	2	16.84	16.14–17.55	0.50	0.71	17.66	17.44	17.55	16.02	16.29	16.14			
M3	APO	4	22.57	21.29–23.36	0.69	0.83	23.57	23.14	23.36	21.33	21.27	21.29	22.36	23.26	
AT	4	17.18	15.59–18.79	1.35	1.16	16.91	16.66	16.78	15.53	15.65	15.59	17.55	18.79	
PT	4	13.76	12.34–14.91	1.33	1.16	14.97	14.81	14.89	12.35	12.34	12.34	12.91	14.91	
HT	2	9.19	8.98–9.40	0.04	0.21	9.29	9.51	9.40	8.86	9.09	8.98			
Notes.

Approximate measurements are marked by (*).

Specimen (ETMNH 8046) exhibits moderate wear on the teeth of the upper and lower dentition (Figs. 6 and 7). Premolars of the lower and upper dentition exhibit a mostly quadrate, molariform condition with moderate anterior and posterior cingula. Exhibiting a more squared to trapezoidal outline, the P2 bears a longer labial edge relative the lingual edge due to a strong, lingually terminating anterior cingulum along the anterior surface of paracone (Table 1). Paracone and anterior cingulum are merging through wear on the right P2 along the anterolingual moiety of the cusp. However, this merger is incomplete as a shallow furrow still separates the median to lingual moiety of the cusp. Trigon and talon are separated by a moderate to deep median valley. Metacone is conic in outline with the left P2 exhibiting slight merger of the cusp with the posterior cingulum along the posterolignual-most edge of the cusp; a shallow furrow separates the posterior edge of the cusp from the cingulum. Hypocone is worn to merger with the posterior cingulum which reduces labially until its termination along the posterolabial to labial edge of the metacone. An ellipsoid fossette is visible on the anterior moiety of the hypocone on the left P2.

Figure 5 Lateral (A) and buccal (B) view of mandible, ETMNH 8046, of Mylohyus elmorei bearing right p2-m3 and left p3-m3.

Image is in grayscale to prevent morphologies from being obscured due to coloration.

Third and fourth upper premolars of ETMNH 8046 exhibit similar morphology except for the latter being slightly larger. Both exhibit a merger of the protocone and the weak anterior cingulum through wear. Protocone is merging anterolabially with the heavily worn paraconule is whereas the paracone remains separated from either feature by a moderate furrow. Metaconule and hypoconule are positioned anteromedial and posteromedial to the hypocone and metacone, respectively. Both metaconule and hypoconule merge with the hypocone through wear. Ellipsoid to semilunar fossettes are located at the center of the metaconule, hypoconule, and hypocone. Metacone is separated from the remainder of the talon by a very shallow furrow. Posterior cingulum is moderate, trending the entire posterior edge of all but the left P3 which exhibits chipping of the enamel posterior to the metacone. A depression is evident along the posterior cingulum of the left P4 and the anterior cingulum of the M1, which may represent a cavity or pathology.

Upper molars of ETMNH 7279 and ETMNH 8046 exhibit typical tayassuid morphology due to the square to rhombohedral placement of the four primary cusps and strong to moderate anterior and posterior cingula. Left M1 exhibits a continuation of the depression of the P4 at the site of paraconule on the anterior cingulum. Paraconules on both the M1 and M2 (of both ETMNH 7279 and ETMNH 8046) are merging with the paracone through wear, whereas the M3 only exhibits merger of the paraconule with the anterior cingulum. All molars exhibit labial cingula that are weak to moderate along the anterolateral edge of the paracone, within the deep median valleys between the labial cusps, and along the posterolateral surface of the metacone. A single accessory cuspule populates the labial cingulum within the median valley of the M2 and M3, however, the cuspule in the latter is reduced. Both the trigon and talon have been almost completely worn into irregularly-shaped, transversely elongate fossettes in the M1. Talon fossette exhibits a partially separation of the metacone fossette from the rest of talon fossette by a thin remnant of lingual to posteriorly bounding enamel (right) or raised dentin (left) that opens anterolingually. Alternatively, the M2 only exhibits merger of the hypocone and metacone with the hypoconule and the metaconule, respectively. Metaconule is heavily worn in the M3 but remains separated from the metacone and hypocone by shallow furrows. In the M2 the hypoconule is worn flat with the posterior cingulum and is merging with the hypocone. Posterior cingulum is strongly developed and bears a small accessory cuspule at its termination at the posterolabial edge of the metacone. Four distinct cusps and conules are present on the moderately worn hypoconulid complex of the M3 in addition to the hypoconulid. Three of these accessory cuspules are arranged in a transverse row along the posterior boundary of the complex and are merging through wear. The remaining cuspule is positioned directly labial to the hypoconule. Posterior cingulum is reduced to a small but strong shelf positioned posterolingual of the metacone and dominated by two small accessory cuspulids.

Table 2 Measurements (mm) of the lower dentition and mandible of Mylohyus elmorei.

							ETMNH 8046	ETMNH 17219	ETMNH 19281	UF 57280	UR/TRO 412	UF 294749	
		N	X ¯	Range	σ2	Σ	Left	Right	Average			Cast of holotype			
Length p2-m3	1	100.81					100.81	100.81						
Length p2-p4	1	39.94					39.94	39.94						
Length m1-m3	2	60.29	59.84–60.95	0.21	0.46	60.95	60.55	60.75				59.84		
Depth of rami at m1	1	43.98					42.69	43.98						
Depth of rami at m3	1	43.00					42.71	43.00						
p2	APO	3	11.98	11.17–12.78	0.43	0.66		11.17	11.17			11.99			
AT	3	7.87	6.63–9.12	1.04	1.02		6.63	6.63			7.87			
PT	3	8.37	7.88–8.72	0.13	0.36		7.88	7.88			8.52			
p3	APO	2	13.62	13.55–13.69	0.00	0.07	13.47	13.64	13.55	13.5 +			13.69		
AT	3	10.92	10.02–12.70	1.59	1.26	10.09	9.99	10.04	10.26		12.70	10.02		
PT	3	11.22	10.69–11.53	0.14	0.38	10.62	10.75	10.69	10.62		11.53*	11.44		
p4	APO	5	14.67	12.74–16.83	1.69	1.30	14.75	14.70	14.73			14.62	14.44	12.74	
AT	5	12.63	11.98–14.19	0.63	0.79	11.97	11.99	11.98			14.19	12.47	12.29	
PT	5	13.63	12.53–16.22	1.74	1.32	12.94	13.07	13.01			16.22	13.36	13.05	
m1	APO	3	16.49	16.20–16.84	0.07	0.27	16.35	16.52	16.44			16.84	16.20		
AT	3	13.43	13.12–13.72	0.06	0.24	13.62	13.81	13.72			13.12	13.45		
PT	3	14.73	13.86–16.41	1.41	1.19	13.88	13.84	13.86			16.41	13.92		
m2	APO	4	19.52	17.74–21.88	2.55	1.60	18.48	18.38	18.43		20.03	21.88	17.74		
AT	4	15.40	14.30–18.00	2.28	1.51	14.22	14.37	14.30		14.69	18.00	14.60		
PT	4	15.86	14.56–19.00	3.37	1.83	14.58	14.53	14.56		15.28	19.00	14.61		
m3	APO	3	26.22	25.36–27.75	1.18	1.09	25.45	25.26	25.36			27.75	25.55		
AT	3	15.64	13.37–18.97	5.80	2.41	13.32	13.42	13.37			18.97	14.57		
PT	3	14.75	13.02–18.10	5.62	2.37	13.12	13.13	13.13			18.10	13.02		
HT	3	11.24	9.94–13.78	3.22	1.79	10.04	9.97	10.00			13.78	9.94		
Notes.

Approximate measurements are marked by (*) whereas incomplete measurements, due to the posterior portion of enamel being absent, are marked by (+).

Lower premolars of ETMNH 8046 exhibit similar morphology to the upper premolars; however, the p2 is more transversely constricted (Table 2). In total, each premolar displays a fully formed protoconid, metaconid, hypoconid, and entoconid. Protoconid and metaconid of the p2 are distinct, but not fully bifurcated as in the p3—in both ETMNH 8046 and ETMNH 17219—and the p4. Anterior cingula are variable between premolars, being weakly developed but become inflated at the site of the paraconulid in the p3 and p4. Metacone of the p3 and p4 exhibits an ellipsoid to rectangular posterolabial extension that is distinct from the parent cusp in ETMNH 17219. Premolar talonid basins exhibit a metaconulid and hypoconulid that are positioned directly anteromedial and posteromedial of the entoconid and hypoconid, respectively, in a ‘cross-’ or ‘plus-’ shaped configuration. Posterior cingulum exhibits additional crenulation and very small accessory cuspules on the p4 that are not present on the p3. A slight elevation of the trigonid cusps, relative to the talonid cusps, is evident in ETMNH 17219 whereas ETMNH 8046 lacks this feature due to a greater degree of wear.

Heavily worn, the m1 of ETMNH 8046 exhibits complete obliteration of all cusps and conules. Trigonid and the anterior cingulum are worn to a single transversely trending fossette. Wear of the talonid produces an ellipsoid fossette with ellipsoid extensions into the positions of the entoconulid and hypoconulid. Trigonid and talonid fossettes are separated from one another by a thin band of enamel on the right m1. However, the left m1 exhibits merger of the trigonid and talonid fossettes in tandem with the posterior cingulum almost being completely worn. Moderately deep, semispherical concavities are present within the dentin at the positions of the metaconid and entoconid of the right m1 and the posterior margin of the posterior cingulum of the left m1 indicating a potential pathology.

Figure 6 Upper dentitions of Gray Fossil Site and Palmetto Fauna Mylohyus elmorei.

Specimens observed include UF/TRO 440 (A), UF 203540 (B), UF 12265 (C), ETMNH 8046 (D), and ETMNH 7279 (E). Image is in grayscale to prevent morphologies from being obscured due to coloration.

Figure 7 Lower dentition of Mylohyus elmorei.

Observed specimens are ETMNH 17219 (A), ETMNH 19281 (B), ETMNH 8046 (C), UF/TRO 412 (D), and UF 294749 (E). Image is in grayscale to prevent morphologies from being obscured due to coloration.

Similar to the m1 in general outline and apparent cusp arrangement, the m2 and m3 are less worn. Anterior cingulum is moderate to strong in both ETMNH 8046 and ETMNH 19281. Angular wear facets along the surface of the protoconid and metaconid merge the cusps anteriorly with the anterior cingulum. Both cusps exhibit a central fossette along the occlusal surface, with the protoconid exhibiting an anterolabial extension of the fossette into the median of the anterior cingulum. Posterolateral projection of the metaconid is merged with the main body of the metaconid through wear in both ETMNH 8046 and ETMNH 19281. Hypoconid is separate from the hypoconulid in ETMNH 19281, but is merged through wear in ETMNH 8046. Hypoconulid is separate in both specimens, however, it is merged with the strong posterior cingulum through wear in ETMNH 8046. Despite being slightly less worn, the m3 exhibits a similar positioning of the primary cusps as the m2 with the presence of a hypoconulid complex. Four distinct cusps or conulids, including the hypoconulid, are positioned on the hypoconulid complex of the right m3; whereas the left m3 exhibits five conules in ETMNH 8046. On both m3′s the hypoconulid is positioned posteromedian of the talonid with the accessory cuspules, of variable size and profile, being positioned posterior to the hypoconulid in a circular arrangement.

Comparisons—Material from the GFS is referred to Mylohyus due to the presence of distinct apomorphies; specifically, a long diastema that exceeds the length of the cheek tooth row and fully molarized premolars (Wright, 1991; Wright, 1998). This material is referred to M. elmorei on the grounds that it bears notable similarity to material previously collected from the Palmetto fauna of Florida (Table S1). Dental morphologies of ETMNH 8046 and a cast of the holotype, MCZ 3805 (labeled UF 57280), display very few differences outside of the latter exhibiting relatively larger dental dimensions for all but the p4. This is due to the holotype having less robust cingula—both anterior and posterior—on the p4 than that of ETMNH 8046. A concave depression on the lingual surface of the m1 metaconid is also evident in the holotype cast. Other potential differences may be obscured due to the p3 missing its talonid on the holotype. Alternatively, the p3 of UF/TRO 412 displays variation of the talonid: with the entoconulid and hypoconulid being merged into a single anteroposterior trending rectangular cuspule that is positioned median of the entoconid and hypoconid. Distinct posterolabial extensions of the metaconid on the p3 and p4 are only present in UF/TRO 412, UF 293749, and UF 57280. Moderate wear obscures the number of accessory cuspules present on the posterior cingula of the p4 in ETMNH 8046, UF/TRO 412, and UF 57280. Five accessory cuspules of variable size are visible on the posterior cingulum of the p4 on UF 294729. Hypoconulid complex exhibits a substantial amount of variation between the observed specimens. Three cuspules (including the hypoconulid) are evident on UF/TRO 412, whereas UF 57280 and ETMNH 8046 exhibit hypoconulid complexes comprised of four and five, respectively.

Dental and cranial morphologies are relatively similar between ETMNH 8046 and UF 12265. Both ETMNH 8046 and UF 12265 exhibit a very elongate post canine diastema, the former being ∼109% of the cheek tooth row, whereas the latter exhibits a diastema of only ∼101% (Fig. 8). Origination of the triangular zygoma is positioned directly dorsal to the P2 in both specimens; however, further comparisons are not possible due to the fragmentary nature of ETMNH 8046. Little deviation outside of tooth dimensions of the P2-M2 is present between ETMNH 8046 and UF 12265. Consequently, the M3 exhibits a greater degree of variation with ETMNH 8046 exhibiting a wider talon and hypoconule complex, relative to the trigon, than those of UF 12265, UF 203540, and UF/TRO 440. Further differences between the specimens are evident in the lateral flaring of the canine buttress in UF 12265 that is not present in ETMNH 8046 (Fig. 9). Specifically, the flaring in UF 12265 begins on the approximate anteroposterior midpoint of the postcanine diastema, whereas the buttresses flare develops within the anterior-most portion of the postcanine diastema in ETMNH 8046. These dental and cranial differences could be indicative of these specimens representing different species; however, this seems premature because the GFS specimen will be the second partial cranium reported for the species. As such, the variation (geographic, temporal, sexual, or individual) present within the species is unknown, however comparisons could be made to known degree of intraspecific variation in Mylohyus fossilis and extant species to show that this range of variation is more likely intraspecific rather than interspecific.

Figure 8 Partial crania, ETMNH 8046 (A) and UF 12265 (B), of Mylohyus elmorei in occlusal view.

Image is in grayscale to prevent morphologies from being obscured due to coloration.

Figure 9 Partial crania, ETMNH 8046 (A) and UF 12265 (B), of Mylohyus elmorei in dorsal view.

Image is in grayscale to prevent morphologies from being obscured due to coloration.

cf. Mylohyus elmoreiWhite, 1942; Wright & Webb, 1984	

Referred Specimens—ETMNH 6767: partial L. zygoma and maxillary fossa.

Description— Highly sinuous cortical bone is evident in both the proximal and distal reconstructions of the left zygomatic wing, ETMNH 6767. Comprised of three associated portions of a left zygomatic wing, ETMNH 6767 exhibits the squamosal portion of zygoma with an intact mandibular fossa. A lambdoid crest of moderate depth projects posterodorsally from the confluence of the zygomatic wing and jugal bar. Ventrolaterally oriented, the semilunar mandibular fossa is positioned posterior and ventral to the zygoma, with the concave-most margin of the fossa being approximately equal in level with the zygoma. Zygoma remnants exhibit a triangular dorsal outline of the posterior margin where the inflated portion reduces posterodorsally to a thin edge. Deeply incised cortical bone demarcates the partial, circular to ellipsoid, rostral muscle fossa on the ventral surface of the reconstructed segments of the main distal body of the zygoma. Another muscle attachment is evident along the flat dorsal surface of this distal section in the form of an anteromedially trending muscle scar, comprised of an elongate raised ridge. It should be noted that both ETMNH 6767 and ETMNH 8046 could potentially be a single individual; however, given the spatial distribution of the two specimens, they are considered separate for this analysis.

Comparisons—Despite being comprised of reconstructed fragments, ETMNH 6767 appears to exhibit affinities to UF 12265 due to the mandibular fossa being positioned posterior and ventral to the posterior margin of the triangular zygomatic wing. This separates ETMNH 6767 from either Protherohyus brachydontus and Prosthennops serus, which exhibit a mandibular fossa that is positioned directly ventral of the trailing edge of the wing-like zygoma. Moreover, the anterolateral to posteromedial angle of the posterior margin of ETMNH 6767 further mirrors UF 122665. Conclusive assignment of ETMNH 6767 is withheld as a larger and less fragmentary sample of M. elmorei is needed for a reliable taxonomic assignment.

Genus ProsthennopsGidley, 1904	
Prosthennops serus (Cope, 1877) Gidley, 1904	

Holotype—AMNH 8511: partial mandible with R. i1,2, p2-m1 and L. i1-3, p2-m3.

Referred Specimens—ETMNH 410: isolated L. p4; ETMNH 5615: partial mandible with L. and R. i1-m3; UF/TRO 413: L. m2-m3; UF 220251: L. m3. Additional referred specimens listed in Hulbert, Morgan & Kerner (2009a) and Schultz & Martin (1975) (Table S1).

Figure 10 Comparison of partial Prosthennops cf. P. serus and Prosthennops serus mandibles, UF 212306 (A and C) and ETMNH 5615 (B and D), respectively.

Image is in grayscale to prevent morphologies from being obscured due to coloration.

Description—As a partial mandible, ETMNH 5615 (Fig. 10) is lacking the coronoid, condylar, and angular processes. Mandibular symphysis is long with the dorsal surface exhibiting a moderately deep spout-like concavity with a posteroventral orientation. This mandibular concavity is bounded by raised ridges of cortical bone that trend the length of the postcanine diastema. Projecting posteriorly, the post-canine diastema is moderate in length—approximately 62.4% of total cheek tooth series length (Table 3). Paired genial pits are positioned along the posterior margin of the symphysis where rami merge to form the symphysis. A single mental foramen is located along the anteroventral surface of the symphysis posteroventral of the i2 along both rami. Another set of foramina are evident along the postcanine diastema with the left bearing three foramina and the right bearing two foramina. Rami laterally broaden in a posterodorsal trend beginning ventral of p3 before being level with the base of the m3. Posterior extent of the broadening appears to be evident along the labial edge of the left m3, however, due to this region being heavily reconstructed this is tentative. Coracoid process originates directly posterior of the m3. Angle appears to originate ventral of the m3; however, the reconstruction of this portion of the rami may be skewing this observation. Submandibular fossa is lacking in much of the specimen; only being evident at the posterior of the rami where it transitions into the shallow pterygoid fossa along the medial surface of the angle ventral to the posterior of the m3. All cheek teeth are bunodont and brachydont.

Table 3 Measurements (mm) of the lower dentition and mandible of Prosthennops serus.

						ETMNH 5615	UF 166243	UF 212306	UNSM 76052T	UNSM 76504T	KUMP 3755H	C.I.T 610C	
	N	X ¯	Range	σ2	σ	Left	Right	Average	Type Cast (AMNH8511)						
Length p2-m3	5	97.71	91.60–103.06	22.86	4.78	101.95	104.16	103.06		102.41*	91.60	92.70	98.80		
Length p2-p4	7	38.23	36.00–40.27	1.95	1.40	37.39	39.15	38.27	38.41	40.27	36.00	36.50	39.20	39.00	
Length m1-m3	5	59.42	53.50–15.98	4.00	4.16	64.22	65.18	64.70		62.50	53.50	56.70	59.70		
Postcanine Diastema	6	54.87	49.90–62.40	28.33	5.32	61.17	63.62	62.40		62.04*	51.50	49.90	53.40	50.00	
% Length of PCD/Length p2-m3	2	60.18	59.82–60.55	0.27	0.52	59.99	61.08	60.55		59.82					
Precanine Diastema	3	6.75	6.26–7.50	0.29	0.54	6.01	6.51	6.26			7.50	6.50			
Length Mandibular symphysis	3	86.14	82.00–90.28	17.13	4.14		92.33			90.28				82.00	
Distance between p2 and symphysis	1						10.66								
Depth of rami at m3	1	39.22								39.22					
Depth of rami at m1	3	45.79	42.27–50.02	10.25	3.20	48.72	51.32	50.02		42.27	45.10				
Width of rami at m3	1	24.91								24.91					
Notes.

Approximate measurements are marked by (*).

Additional measurement data is taken from (T) Schultz & Martin (1975), (H) Hesse (1935), and (C) Colbert (1938).

Anterodorsally oriented, the incisors of ETMNH 5615 exhibit a subspatulate to subconical morphology with the i3 exhibiting a reduced peg-like condition (Table 4). All incisors are procumbent. Wear is evident on the occlusal surface of the i1 in the development of a pseudo-cylindrical wear facet comprised of an ellipsoid fossette bound by enamel. The i2 is worn, with the anterior-most portion equal to the surface of the wear facet for the i1. Remaining portions of the elliptical wear facet exhibit a posterolabial trend. Peg-like i3, only present on the right, appears to exhibit rounding of its occlusal surface. A small diastema—approximately 7 mm—occurs between the i3 and anterior boundary of the canine. Canines are typical of tayassuids, with a triangular outline and occlusal wear on the posterior surface.

Table 4 Measurements (mm) of the lower dentition and mandible of Prosthennops serus.

							ETMNH 5615	ETMNH 410	UF 166243	
											Type Cast	
		N	X ¯	Range	σ2	σ	Left	Right	Average		(AMNH8511)	
i1	APO	1	8.40		0.00	0.00	8.37	8.43	8.40			
AT	2	5.90	4.50–7.30	1.96	1.40	7.06	7.55	7.30			
PT	0	 									
i2	APO	3	9.94	5.41–12.70	10.42	3.23		5.41	5.41			
AT	4	6.59	5.00–8.56	1.80	1.34	8.41	8.71	8.56			
PT	0	 									
i3	APO	2	5.11	4.70–5.51	0.16	0.40		5.51	5.51			
AT	3	3.19	2.50–4.47	0.82	0.90		4.47	4.47			
PT	0	 									
Canine	APA	5	16.33	13.74–18.40	3.11	1.76	13.74		13.74			
Transverse	4	13.21	12.24–14.60	0.86	0.93	11.91	12.57*	12.24			
p2	APO	7	10.79	9.92–11.25	0.20	0.45	9.79	10.06	9.92		10.93	
AT	6	6.93	6.39–7.60	0.17	0.41	6.31	6.46	6.39		6.54	
PT	3	7.25	7.16–7.33	0.01	0.07	7.19	7.37	7.28		7.16	
p3	APO	7	12.49	11.80–13.00	0.12	0.35	12.77	12.53	12.65		12.48	
AT	6	9.34	9.00–9.70	0.06	0.25	9.01	9.26	9.14		9.22	
PT	3	9.41	9.31–9.61	0.02	0.14	9.48	9.73	9.61		9.33	
p4	APO	8	15.08	13.20–17.36	1.36	1.16	15.75	16.14	15.94	17.36	15.00	
AT	7	11.97	11.07–12.64	0.21	0.46	11.57	12.00	11.78	11.07	11.86	
PT	4	13.10	12.40–13.85	0.30	0.55	13.17	13.55	13.36	12.78	12.40	
m1	APO	7	14.83	13.70–15.98	0.56	0.75	14.94	14.98*	14.96		15.38	
AT	6	12.61	12.00–13.20	0.19	0.43	12.96	12.78	12.87		12.10	
PT	3	13.04	12.19–13.96	0.53	0.73		12.97	12.97		12.19	
m2	APO	8	18.82	16.80–20.77	1.35	1.16	20.51	21.02	20.77		19.71	
AT	7	14.98	14.00–16.00	0.46	0.68	16.00	15.99	16.00		15.51	
PT	4	15.84	15.42–16.40	0.12	0.35	16.25	16.54	16.40		15.42	
m3	APO	7	27.61	24.50–32.26	5.92	2.43	29.49	29.77	29.63			
AT	7	16.14	14.90–18.52	1.32	1.15	16.60	16.73	16.67		15.51	
PT	5	16.09	14.86–17.60	0.95	0.97	16.40	16.77	16.58		14.86	
HT	4	12.61	12.29–13.14	0.10	0.32	12.56	12.53	12.54			
		UF 220251	UF 212306	UF/TRO 413	UNSM 76052T	UNSM 76504T	KUMP 3755H	C.I.T 610C	
i1	APO								
AT							4.50	
PT								
i2	APO				12.70	11.70			
AT				7.00	5.80		5.00	
PT								
i3	APO					4.70			
AT					2.60		2.50	
PT								
Canine	APA				18.40*	16.50	18.00	15.00	
Transverse				14.60*	12.50		13.50	
p2	APO		11.25		11.00	10.30	11.10	11.00	
AT		7.26		7.60	6.80		7.00	
PT		7.33						
p3	APO		13.00		11.80	12.70	12.30	12.50	
AT		9.39		9.70	9.60		9.00	
PT		9.31						
p4	APO		15.62*		13.20	14.70	14.30	14.50	
AT		12.64		12.40	12.00		12.00	
PT		13.85						
m1	APO		15.98*		15.30	13.70	14.00	14.50	
AT		12.89		13.20	12.60*		12.00	
PT		13.96						
m2	APO		19.23	18.73	16.80	18.60	19.20	17.50	
AT		14.84	15.58	14.40	14.50		14.00	
PT		15.74	15.80					
m3	APO	32.26	28.14	26.57	24.50	25.80	26.40		
AT	18.52	16.32	16.03	14.90	15.00			
PT	17.60	16.17	15.25					
HT	13.14	12.48	12.29					
Notes.

Approximate measurements are marked by (*).

Additional measurement data is taken from (T) Schultz & Martin (1975), (H) Hesse (1935), and (C) Colbert (1938).

Triangular in occlusal outline, the p2 of ETMNH 5615 exhibits two roots (Fig. 11). Protoconid, conic in profile, is the primary cusp of the p2 and is well elevated above the talonid cusp/cuspule. A weak anterior cingulum trends along the anterior surface of the protoconid. Talonid cusp/cuspule is positioned directly posterolabial of the protoconid on the right p2. Left p2 appears to lack this cusp due to damage and/or merger through wear. A weak lingual cingulum trends from the posterolabial edge of the protoconid along the labial and posterior edges of the talonid basin.

Figure 11 Comparison of lower dentition of Prosthennops serus and Prosthennops cf. P. serus.

Observed specimens: ETMNH 5615 (A), UF 212306 (B), UF/TRO 413 (C), and UF 220251 (D). Image is in grayscale to prevent morphologies from being obscured due to coloration.

Trapezoidal in occlusal outline, the p3 of ETMNH 5615 exhibits four primary cusps and evidence for two to three roots. Trigonid is comprised of poorly bifurcated protoconid and metaconid that may merge with wear. A strong anterior cingulum is positioned along the anterior base of the trigonid cusps. An accessory cuspule, or posterolabial extension of the metaconid, is evident along the posterior margin of the protoconid and metaconid. Merger of this feature with the metaconid through wear is present in ETMNH 5615. Talonid is comprised of two rounded cusps/cuspules that are separated from the trigonid cusps (and themselves) by weak valleys. A weak labial cingulum extends across the short valley between the protoconid and the hypoconid. Evidence is present for a posterior cingulum but both left and right p3 exhibit an elongate fossette and/or damage along the posterior margin of the tooth.

Similar in cusp morphology to the p3, the p4 exhibits a more quadrate condition and four roots. Cusps are subequal in height and conic with the trigonid cusps being elevated dorsal to the talonid cusps. Weakly to moderately worn in nature, ETMNH 410 exhibits a moderate anterior cingulum along the base of the trigonid. Anterior cingulum of ETMNH 5615 merges with the trigonid through wear. Metaconid is merging with its posterior extension or accessory cuspule in both ETMNH 410 and ETMNH 5615. Deep valleys separate the trigonid and talonid, with the labial valley exhibiting a moderate cingulum between the posterior edge of the protoconid and the anterior edge of the hypoconid. Entoconulid does not appear to be present in either ETMNH 410 or ETMNH 5615. Hypoconid exhibits anterolingual extension of its wear facet in ETMNH 410 toward where the entoconulid would be positioned in a molariform premolar, however, no evidence of a distinct cusp is present. In ETMNH 5615 the hypoconid and entoconid wear to a circular occlusal profile, with centrally positioned circular to ellipsoid fossettes. Hypoconulid remains separate in both specimens being positioned along the posterolingual edge of the hypoconid.

Despite being heavily worn in ETMNH 5615, the m1 exhibits quadrate, four rooted condition. Enamel is only present along the lingual edge of the right m1 and along the entire labial edge and lingual edge of the metacone of the left m1 due to the entoconid being absent. Trigonid and talonid fossettes are transversally ellipsoid and completely merged. Anterior margin of both the right and left m1′s of ETMNH 5615 exhibit a concave depression that conforms to the posterior margin of the preceding p4.

Heavy wear on the m2′s of ETMNH 5615 is evident as the protoconid and metaconid on both teeth are worn to low mounds that are merging at the median valley, now reduced to a very weak furrow. Protoconid exhibits an ellipsoid fossette with an extension to the paraconule and anterior cingulum. Metaconid also exhibits an ellipsoid fossette bearing an ellipsoid extension into the merged posterior extension or accessory cuspule. Trigonid and talonid are still separated by a moderate valley that is weak anterolingual of the entoconulid. A weak labial cingulum is present between the protoconid and hypoconid. Hypoconid is merging anterolingually with the entoconulid, as well as posterolingually with the hypoconulid and posterior cingulum. A circular fossette dominates the center of the hypoconid with an ellipsoid extension into the site of the entoconulid. Hypoconulid, despite being worn flat to the posterior cingulum, exhibits an ellipsoid fossette that remains separate from the hypoconid fossette. Entoconid remains separate with an ellipsoid fossette dominating the center of the cusp.

Cusps of the m3 of ETMNH 5615 exhibit angular wear along the anterior and posterior surface with the metaconid and entoconid exhibiting less wear. A strong anterior cingulum trends across the anterior of the trigonid cusps. Protoconid exhibits anterolingual merger with the paraconulid and anterior cingulum. Metaconid is merged with its posterolabial extension or accessory cuspule. Deep valleys separate the trigonid and talonid, while a labial cingulum trends between the protoconid and hypoconid. Hypoconulid is merged anterolingually with the entoconulid but remains separate from the hypoconulid. Moreover, the m3 exhibits a bulbous hypoconulid complex with a single broad and robust cusp posterior to the anteroposteriorly compressed hypoconulid. Left m3 exhibits merger of the hypoconulid with the heel cusp along the posterolabial edge of the cuspule.

Comparisons—Specimens are attributed to Prosthennops serus due to the presence of a robust, bunodont and brachydont dentition. At the generic level, these specimens can be differentiated from Mylohyus based on the presence of submolariform p2′s and a post-canine diastema that is less than the length of the cheek tooth row. Moreover, all specimens being assigned to Prosthennops serus exhibit a triangular p2 with a single prominent cusp anterior to the talonid; further distinguishing these specimens from the lophate, subzygodont to zygodont Protherohyus brachydontu s and Platygonus pollenae. Cranial apomorphies specific to Prosthennops serus (e.g., distally angular zygomatic wings, zygoma originating dorsal to premolars (Wright, 1991; Wright, 1998)) are not evident in the GFS material due to the lack of crania.

Partial mandible, ETMNH 5615, is comparable to UF 212306, UF 166243, a cast of the type specimen (AMNH 8511), originally described by Cope (1877), UNSM 76052, UNSM 76054, and UNSM 76059 (Table S1). Symphyses of these specimens exhibit a deep anteroposteriorly trending semi-cylindrical spout-like concavity that opens along the posterior margin of the symphysis. Moreover, each of these specimens’ exhibit dentitions that are bunodont and brachydont with a submolariform p2 and p3 and a molariform p4. Dentition of UF 212306 and UF 166243 is less worn compared to ETMNH 5615 (Fig. 11). Continuation of the anterior cingulum along the anterolabial edge of the protoconid into the labial cingulum on the p3 differentiates UF 212306, ETMNH 5615, UF 166243, UNSM 76052, and UNSM 76054. In ETMNH 5615, UF 166243, UNSM 76052, and UNSM 76054 this cingulum terminates along the anterior edge of the protoconid, with an isolated labial cingulum present within the median valley between the protoconid and hypoconid. Labial cingulum trends posteriorly along the labial edge of the hypoconid in UF 166243. A labial cingulum is also observed in the p4 with ETMNH 5615, UF 212306, UNSM 76052, UNSM 76054, and UNSM 76059 exhibiting a labial cingulum that is restricted within the median valley. Alternatively, UF 166243 exhibits an extension of the cingulum along the labial to posterior margin of the hypoconid of the p4. None of the observed specimens adequately represent the cusps of the m1 due to wear or loss through damage. Remaining molars exhibit a similar morphology, with ETMNH 5615 exhibiting more labiolingually broad and robust anterior and posterior cingula of the m2 and m3 than UF 212306, UF/TRO 413, UF 166243, UNSM 76052, UNSM 76054, or UNSM 76059. Both UF 220251 and UF/TRO 413 exhibit m3′s that are comparable to ETMNH 5615, UF 212306, UNSM 76052, UNSM 76054, and UNSM 76059 due to the relative morphology of the cusps and the hypoconulid complex being dominated by two to three poorly bifurcated accessory cusps that may merge together through wear into a single prominent cusp. Overall, ETMNH 5615, UF 212306, UF220251, and UF/TRO 413 exhibit similar dental and mandibular characteristics to the cast of the type specimen—UF 166243—and those described in Hesse (1935), Colbert (1938), and Schultz & Martin (1975); however, dental dimensions vary within the sample (Fig. 11). Specimens from the GFS and Tyner Farm Locality are proportionally larger than the material described by Hesse (1935), Colbert (1938), and Schultz & Martin (1975), indicating greater interspecific variation than previously recognized.

Discussion

Previously only known from the Palmetto Fauna of Florida (Wright & Webb, 1984; Wright, 1991; Wright, 1998) within the Fort Green Mine, Palmetto Mine, Payne Creek, Saddle Creek Mine, and South Pierce quarries, M. elmorei exhibits a northward expansion into the Appalachian region with the inclusion of GFS material (Fig. 1). Although this discovery expands the range of M. elmorei, it does not negate the assertion by Webb et al. (2008) that the species is endemic to the southeastern North America. Mylohyus, as a genus, is widespread throughout parts of North America (Wright, 1998). Mylohyus fossilis in particular is prevalent through the Blancan to Rancholabrean of the central and southeastern regions of North America (Kinsey, 1974; Kurten & Anderson, 1980; Wright, 1991; Wright, 1998). Alternatively, another Hemphillian species within the genus, M. longirostris, is reported from the John Day region based on a single rami and fragmentary cranial material (Thorpe, 1924; Wright, 1991; Wright, 1998). Wright (1998) attributes material collected from the Hemphillian Mixon’s Bone Bed local fauna of Florida as being affiliated to M. longirostris; however, only the locality is designated. No specimen data is reported to verify this record. In sum, the geographic distribution of these species illustrates the potential for a larger distribution for M. elmorei, however, the rarity of the species within given localities can make further range expansions difficult to determine.

Confirmation of Prosthennops serus at the GFS expands the known range of the taxon eastward and northward into the Appalachian Mountain region, making GFS the second eastern-most locality from which Prosthennops serus is recognized (Fig. 1). Following Wright (1998), Prosthennops serus—senso stricto—is known from the early Clarendonian of Kansas (Cope, 1877; Wright, 1998), earliest Hemphillian of Oregon (Colbert, 1938), earliest to late early Hemphillian of Nebraska (Hesse, 1935; Schultz & Martin, 1975), earliest Hemphillian to Blancan of an unnamed unit within Hidalgo, Mexico (Wright, 1998), late early Hemphillian of Alabama (Hulbert & Whitmore, 2006), and early Hemphillian Tyner Farm locality of Florida (Hulbert, Morgan & Kerner, 2009a). Material from the late to latest Hemphillian of the Coffee Ranch Fauna of Texas—approximately 6.6 Ma (Passey et al., 2002)—and Ocote Fauna of Mexico are also referred by Wright (1998), however, no catalog numbers are listed resulting in ambiguity regarding whether this refers to new or reassigned material. Other localities listed by Wright (1998) as bearing material comparable to Prosthennops serus are located within the earliest Hemphillian of the Deer Lodge Basin of Montana, late early Hemphillian Higgins Local Fauna of Texas, and late early Hemphillian of the Wray Fauna of Colorado. Webb & Perrigo (1984) also refer a well-worn m3 as being comparable to the species from the Gracias Fm. of Honduras, however, the worn nature of the tooth and predominant use of anteroposterior and transverse measurements make this identification suspect.

Overall, the presence of M. elmorei and Prosthennops serus within the fauna of the GFS provides further evidence for a forested environment. Additionally, their presence draws further parallels to the Palmetto Fauna. Mylohyus elmorei and Protherohyus brachydontus within the Palmetto Fauna are referred to by Webb et al. (2008) as browse-dominated mixed-feeders. DeSantis & Wallace (2008) report that two of the GFS tayassuid specimens exhibit a C 3 dominated dietary profile. Despite the specimens being attributed to M. elmorei and Prosthennops serus not being recovered until after DeSantis & Wallace (2008), a similar browsing diet is suggested for the GFS M. elmorei and Prosthennops serus material based on morphology. Specifically, the presence of a bunodont and brachydont dentition is cited by Hulbert (2001) as an indicator for M. fossilis being a forest species that subsisted on fruit, nuts, and succulents. Additional evidence for this dietary preference in fossil peccaries, including M. elmorei, has been reported using stable carbon isotope and dental microwear texture analyses (Yann & DeSantis, 2014; Bradham et al., 2018). Similar parallels are drawn by Woodburne (1968), Kiltie (1981), Sowls (1997) and Wright (1998) based on observations on the populations of modern woodland populations of Pecari (=Dicotyles) tajacu and Tayassu pecari. Additionally, potential dental pathologies (e.g., caries such as those described by Andrews (1973), Coyler (1990), Figueirido et al. (2017), and Wang et al. (2017)) on the m1′s of ETMNH 8046 further suggest a frugivorous or sugar-rich diet that would fit in with the current interpretation of the site being an oak-hickory forest.

Recent assessment of the mammalian fauna suggests an age of 4.9 to 4.5 Ma for the GFS (Samuels, Bredehoeft & Wallace, 2018). This suggestion brings the maximum age of the GFS to be in line with that of the Palmetto Fauna of Florida, which is interpreted to be 5.0–4.5 Ma (Tedford et al., 2004; Webb et al., 2008). Presence of M. elmorei could be used to reinforce the upper age limit of GFS, however, there is a possibility that the GFS represents an earlier or later record for M. elmorei. Moreover, the presence of Prosthennops serus at the GFS cannot be utilized to constrain the age due to the species being known from the latest Clarendonian to earliest Blancan (Wright, 1998). Further verification of material from other sites and radiometric analyses, where permissible, are needed to utilize any of the GFS tayassuids for further constraining the site’s biochronology.

Conclusions

Within the GFS tayassuid material a total of two individuals attributed to M. elmorei and Prosthennops serus, respectively, are recognized through systematic analyses. Accordingly, the known distribution of M. elmorei and Prosthennops serus is expanded north into the Appalachian region; the first reported instance of M. elmorei outside the Palmetto Fauna of Florida. Moreover, the presence of M. elmorei emphasizes further parallels between the Palmetto Fauna and the GFS reinforcing the paleoenvironmental interpretation of the latter and suggesting a greater connectivity between the faunas than previously thought. Indeterminant tayassuid material that cannot be directly assigned to either species is evident within the GFS fauna, however, the limited and fragmentary nature of the remaining tayassuid material prevents the designation of another species at this time. Future work focused on this material, in particular the postcranial material, is necessary to further discern the ecology and morphological variation of these species both within the GFS and late Hemphillian to early Blancan of North America.

Supplemental Information

Table S1 Specimens of Mylohyus elmorei and Prosthennops serus referenced in the text

All specimens are listed based on their locality and the publication from which they were initially described.

Click here for additional data file.

We thank Jim Mead for comments and suggestions on an earlier version of this manuscript. Moreover, further thanks are due to Sandra Swift for her aid and overall support throughout the project. Additional thanks to Shawn Haugrud, Brian Compton, April Nye, and Anthony Woodward for their assistance in ETSU collections. We thank Dr. Richard Hulbert for providing access to the University of Florida (UF) collections and providing references and feedback related to the collection. Recognition for the photography goes to Sean Moran. Express thanks are also due to Dr. James Farlow and Ronald Richards for providing access and consultation regarding the tayassuid material associated with the Pipe Creek paleosinkhole. We thank Dr. Hugh “Greg” McDonald for his assistance in clarifying morphological nomenclature. Lastly, we thank reviewers for their helpful comments and edits to this manuscript.

Additional Information and Declarations

Competing Interests

Author Contributions

Data Availability

The authors declare there are no competing interests.

Evan M. Doughty conceived and designed the experiments, performed the experiments, analyzed the data, contributed reagents/materials/analysis tools, prepared figures and/or tables, authored or reviewed drafts of the paper, approved the final draft.

Steven C. Wallace and Blaine W. Schubert contributed reagents/materials/analysis tools, authored or reviewed drafts of the paper, approved the final draft.

Lauren M. Lyon contributed reagents/materials/analysis tools, prepared figures and/or tables, authored or reviewed drafts of the paper, approved the final draft.

The following information was supplied regarding data availability:

Measurement data appears in Tables 1 to 4. Additional specimens are in Table S1.

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
