# Peer review of "First occurrence of the enigmatic peccaries Mylohyus elmorei and Prosthennops serus from the Appalachians: latest Hemphillian to Early Blancan of Gray Fossil Site, Tennessee"

_PeerJ, doi:10.7717/peerj.5926_

## Round 0.1 · original submission · Minor Revisions

The comments from the reviewers are comparatively minor, and hopefully should be easy to incorporate. Reviewer 1's notes on the issue of niche partitioning is perhaps the most critical--this should be addressed in your revised manuscript.

·

Basic reporting

Overall the paper is well written, the identifications of the two species appear to be correct, the descriptions are good if perhaps a little too detailed, and the specimen images and tables are adequate. There are a few grammatical and formatting errors.

Experimental design

The weakest part of the paper is the paleoecology section that runs from the bottom of page 26 to the top part of page 28. The abstract states that “The preservation of these two species from GFS suggests tayassuid niche partitioning in this ancient forested ecosystem.” The authors do not provide any direct evidence of niche partitioning between the two GFS peccaries (e.g., microwear or stable C isotope analysis of paleodiet; biomechanical analysis of jaw mechanics, etc.). Basically, it is just the two species are both present at the site, therefore niche partitioning. Furthermore, the authors have not presented evidence that the two species actually co-occur in the same beds or levels from the GFS, meaning that they might not be truly sympatric in the ecological sense. Recent papers by Yann and DeSantis (2014) and Bedham et al. (2018) that do these types of studies are not cited. I would recommend that this section either be enhanced by further analysis of the paleoecology of the two species at the GFS and expand this section, or delete it altogether.

Validity of the findings

other than the point made in #2, the findings and and analyses are valid

Reviewer 2 ·

Basic reporting

In the Introduction section the abbreviated GFS should be placed first as Gray Fossil Site, then subsequently as GFS.

In Figure 1, if you choose to follow the taxonomy suggested by Parisi Dutra 2017, use Protherohyus instead of Catagonus.

At line 355, change "peg like" to "pig-like"

Experimental design

No comment

Validity of the findings

no comments

---

## Round 0.2 · accepted · Accept

Thank you for your close attention to the comments from the reviewers.

#